# Pregnancy Recommendations Solely Based on Preclinical Evidence Should Be Integrated with Real-World Evidence: A Disproportionality Analysis of Certolizumab and Other TNF-Alpha Inhibitors Used in Pregnant Patients with Psoriasis

**DOI:** 10.3390/ph17070904

**Published:** 2024-07-07

**Authors:** Mario Gaio, Maria Giovanna Vastarella, Maria Giuseppa Sullo, Cristina Scavone, Consiglia Riccardi, Maria Rosaria Campitiello, Liberata Sportiello, Concetta Rafaniello

**Affiliations:** 1Campania Regional Centre for Pharmacovigilance and Pharmacoepidemiology, 80138 Naples, Italy; cristina.scavone@unicampania.it (C.S.); consiglia.riccardi@unicampania.it (C.R.); liberata.sportiello@unicampania.it (L.S.); concetta.rafaniello@unicampania.it (C.R.); 2Section of Pharmacology “L. Donatelli”, Department of Experimental Medicine, University of Campania “Luigi Vanvitelli”, 80138 Naples, Italy; 3Department of Women, Child and General and Special Surgery, University of Campania “Luigi Vanvitelli”, 80138 Naples, Italy; mariagiovannavastarella@hotmail.it; 4AOU Policlinico, Università degli Studi della Campania “L. Vanvitelli”, 80138 Naples, Italy; mariagiuseppa.sullo@policliniconapoli.it; 5Department of Obstetrics and Gynaecology and Physiopathology of Human Reproduction, ASL Salerno, 84124 Salerno, Italy; mr.campitiello@sanita.it

**Keywords:** certolizumab, adalimumab, etanercept, infliximab, golimumab, pregnancy, psoriasis, TNF-alpha, safety, pharmacovigilance

## Abstract

Treatment for pregnant women with psoriasis is limited by the lack of information typically related to clinical trials. While anti-tumor necrosis factor (TNF) drugs offer therapeutic benefits, their safety during pregnancy is a concern. Notably, certolizumab is comparatively safer than adalimumab, etanercept, infliximab, and golimumab according to the current recommendations. Thus, this study aimed to conduct a pharmacovigilance comparative analysis of maternal and neonatal outcomes associated with certolizumab versus other anti-TNF drugs by using data from EudraVigilance. A descriptive analysis was performed of Individual Case Safety Reports (ICSRs) associated with an anti-TNF drug and related to the pregnant patients with psoriasis from 2009 and 2023, focusing our analysis on the specific pregnancy outcomes and fetal/neonatal disorders. The most common pregnancy-related adverse event was spontaneous abortion, predominantly related to adalimumab and certolizumab. Certolizumab was also reported in cases of caesarean section, gestational diabetes, abortion, fetal death, fetal distress syndrome, pre-eclampsia, and premature separation of placenta. Generally, the findings from our study depicted a safety profile that overlapped for each anti-TNF drug, both in maternal/neonatal outcomes and other adverse events, suggesting no substantial differences between treatments. We advocate for further investigations before making concrete recommendations.

## 1. Introduction

Psoriasis is one of the commonest chronic, immune-mediated inflammatory skin condition that causes red, scaly plaques occurring most commonly on the elbows, knees, scalp, and lower back, but it can affect any skin surface [1]. It is increasingly recognized as a systemic illness, with associations documented with psychological, metabolic, arthritic, and cardiovascular complications [2]. Most chronic inflammatory disorders are more prevalent in women than in men, including women of child-bearing age [3]. While there is no consensus regarding gender differences in psoriasis, numerous studies showed a higher frequency in women than in men, while others reported the opposite finding [4]. The observed gender difference in psoriasis incidence may be attributed to the role of sex hormones. These hormones may modulate the biological and immune responses in the skin and can contribute to fluctuations in the activity of psoriasis during particular life periods of women, such as menstruation, menopause, and pregnancy [5]. The exact prevalence of psoriasis in pregnant women is unclear; however, since the majority of patients are diagnosed before the age of 40, the impact of the disease on this population is likely significant [6].

Given the substantial evidence supporting the role of tumor necrosis factor-alpha (TNF-α) in the pathogenesis of several inflammatory conditions, such as psoriasis, anti-TNF drugs including infliximab, etanercept, adalimumab, certolizumab, and golimumab offer an effective therapeutic option that significantly improves the signs and symptoms of psoriasis (along with inflammatory bowel disease, rheumatoid arthritis, ankylosing spondylitis, and psoriatic arthritis) [7,8,9,10,11]. Golimumab is the only anti-TNF drug approved for psoriatic arthritis, but not currently for psoriasis [12].

Regarding female pregnant patients, according to guidelines and recommendations, all anti-TNF drugs can safely be used during the first two trimesters of gestation, particularly etanercept and certolizumab [13,14]. In fact, according to the Summary of Product Characteristic (SmPC), adalimumab, etanercept, infliximab, and golimumab should only be used during pregnancy “if clearly needed”, while certolizumab should only be used during pregnancy “if clinically needed”.

Infliximab, etanercept, adalimumab, and golimumab cross the placenta and have been detected in the serum of infants born to female patients treated during pregnancy, with a consequent increased potential risk of infection, including serious disseminated infection that can become fatal. Certolizumab is the only anti-TNF agent that does not cross the placenta, given that it lacks a fragment of crystallizable (Fc) portion, which is a part of a complete antibody [15].

During pregnancy, antibodies transfer from mother to fetus via the placenta, mainly through the neonatal Fc receptor (FcRn), boosting newborn immunity. Fetal Ig levels rise steadily from the second trimester until delivery, with most transferred in the third trimester. IgG1 is the most efficiently transported subclass. FcRn, expressed by syncytiotrophoblasts, mediates IgG transfer by absorbing maternal IgG into endosomes, where tight binding occurs. Gradual acidification allows for binding, and fusion with the fetal membrane promotes dissociation, ensuring effective transfer (Figure 1) [16].

Preclinical study results support the hypothesis that certolizumab has low placental transfer in pregnant women [17,18,19]. Moreover, a pharmacokinetic study, including pregnant women receiving certolizumab for a locally approved indication, showed a lack of in utero fetal exposure during the third trimester, supporting the continuation of certolizumab treatment during pregnancy [20]. Generally, the dermatology literature contains relatively little safety data, with the majority of anti-TNF drug exposures occurring only during the first trimester. Case reports and small case series have not revealed any adverse pregnancy outcomes in pregnant patients with psoriasis [21,22]. In a small case series, 6 of 12 women with psoriasis received an anti-TNF drug in the first trimester. In this study, there were no congenital anomalies at birth or unfavorable effects on growth and physiological development at the final follow-up [23].

Moreover, psoriasis itself is a risk factor for pregnancy- and neonatal-related outcomes. A recent systematic review found that psoriasis may increase the risk of some pregnancy-related indicators: gestational diabetes, cesarean delivery, pre-eclampsia, gestational hypertension, preterm birth, spontaneous abortion, and some neonatal-related outcomes like low birth weight, small for gestational age, Apgar score < 7, and stillbirth [24].

It is plausible that anti-TNF drugs are associated with an increased risk of infection in patients with psoriasis [25]. Despite this, the risk of maternal and newborn infections is largely studied only for inflammatory bowel disease, and there is no relationship between anti-TNF treatment exposure during pregnancy and an increased risk of maternal and infant infections [26,27,28].

The evidence from both preclinical and clinical studies and the recommendations of guidelines often presents conflicting views. Furthermore, women are frequently underrepresented in pre-approval clinical trials, with pregnant women facing even greater exclusion due to pregnancy often serving as an exclusion criterion in patient enrollment [29]. This underrepresentation not only impacts the generalizability of findings, but also poses significant challenges in understanding the efficacy and safety of interventions across diverse populations. Moreover, in the real-world practice, such disparities in trial demographics can result in notable differences in the use and outcomes of therapies; for instance, the recent European data showed that certolizumab is more commonly prescribed to women with psoriasis outside of clinical trials than within them [30]. This discrepancy highlights the potential limitations in the generalizability of the clinical trial results. Addressing these disparities is important for advancing equitable healthcare and ensuring that new pharmacological treatments are tailored to the needs of all individuals. In this context, real-world data can offer new evidence, particularly regarding specific subpopulations such as pregnant women. In particular, pharmacovigilance databases provide information on the safety of pharmacological treatments used during pregnancy, offering insights that may not be apparent in traditional clinical trials.

The aim of this pharmacovigilance study was to analyze data on safety of anti-TNF drugs indicated for psoriasis in pregnant women. Specifically, we conducted a comparative analysis of individual case safety reports related to the anti-TNF drugs, with a specific focus on pregnant-outcome events and neonatal infections, comparing infliximab, etanercept, adalimumab, and golimumab (all of which can cross the placenta) with the Fc-free certolizumab.

## 2. Results

From 1 January 2009 to 31 December 2023, a total of 340,164 Individual Case Safety Reports (ICSRs) related to anti-TNF drugs were retrieved from EudraVigilance. We excluded 314,146 ICSRs reporting data about patients using an anti-TNF drug for an indication other than psoriasis. Of these, 10,175 ICSRs were related to pregnant patients (3.2%). Finally, we filtered the dataset to include only pregnant female patients, resulting in a subset of 1050 ICSRs for our analysis. Therefore, 10.3% of the pregnant patients treated with an anti-TNF drug had a diagnosis of psoriasis.

The majority of the included ICSRs reported certolizumab as a suspected drug (24.7%), followed by etanercept (22.1%), infliximab (18.8%), adalimumab (17.4%), and golimumab (17.0%) (Figure 2).

For each of the five medications, more than half of the ICSRs included in the analysis lacked patient age information, were spontaneous, and were primarily sourced from patients themselves (Table 1). Additionally, information regarding the ICSRs that included non-pregnant women can be found in Appendix A.

The most frequently reported suspected adverse drug reactions (ADRs) were categorized in the System Organ Class (SOC) “General disorders and administration site conditions” (N = 5505, 22.1%), followed by suspected ADRs categorized as “Musculoskeletal and connective tissue disorders” (N = 4928, 19.8%), “Injury, poisoning and procedural complications” (N = 4294, 17.2%), “Gastrointestinal disorders” (N = 2210, 8.9%), “Skin and subcutaneous tissue disorders” (N = 2114, 8.5%), and “Investigations” (N = 1729, 6.9%). No differences have been observed between different anti-TNF drugs (Figure 3).

The most frequently observed outcome across all five medications was “Not recovered/not resolved”—adalimumab (80.7%), certolizumab (81.8%), etanercept (84.3%), golimumab (84.3%), and infliximab (84.7%)—with relatively few cases showing resolution. Additionally, a notable number of outcomes were categorized as unknown for both anti-TNF drugs (Figure 4).

We measured the reporting odds ratio (ROR) with 95% confidence interval (95% CI) for each SOC, comparing certolizumab with other anti-TNF drugs, both in pregnant and non-pregnant patients. No statistically significant differences were observed between treatments in reporting adverse events classified as cardiac disorders, hepatobiliary disorders, metabolic disorders, neoplasms, reproductive system and breast disorders, respiratory disorders, and vascular disorders, both in pregnant and non-pregnant patients. Moreover, we observed a higher risk of reporting certain types of adverse events (e.g., gastrointestinal disorders, immune system disorders, infections, musculoskeletal disorders, nervous system disorders, skin disorders, and renal disorders) with certolizumab in non-pregnant patients, while the same differences were not observed in pregnant patients. In pregnant women, adverse events classified as congenital, familial, and genetic disorders were more commonly associated with certolizumab in pregnant patients, whereas fewer than three cases were observed in non-pregnant women (Table 2).

We observed significant differences between the results from the main disproportionality analysis—which excluded ICSRs with missing indications—and the results from the sensitivity analysis, particularly in the following SOCs: “Blood and lymphatic system disorders”, “Endocrine disorders”, “Eye disorders”, “General disorders and administration site conditions”, “Immune system disorders”, “Infections and infestations”, “Investigations”, “Metabolism and nutrition disorders”, “Musculoskeletal and connective tissue disorders”, “Nervous system disorders”, “Psychiatric disorders”, “Renal and urinary disorders”, and “Respiratory, thoracic and mediastinal disorders”. Specifically, in the standard disproportionality analysis, we observed greater variations in the reporting of adverse events between pregnant and non-pregnant populations. This suggests that the exclusion of records with unspecified indications may have contributed to a higher perceived disparity in adverse event reporting. The results from the sensitivity analysis are provided in the Appendix A.

In Figure 5, we reported the frequency of suspected adverse drug reactions (ADRs) included in the following Standardized MedDRA Queries (SMQs): “Pregnancy, labor and delivery complications”, “Fetal disorders”, “Neonatal disorders”, and “Termination of pregnancy”. The most common adverse event was “spontaneous abortion”, predominantly related to the use of adalimumab (37.2%) and certolizumab (32.7%). Certolizumab was also reported in cases of caesarean section (12.1%), gestational diabetes (9.1%) and meconium in amniotic fluid (4.5%), abortion (3.0%), fetal death (3.0%), fetal distress syndrome (3.0%), pre-eclampsia (3.0%), premature separation of placenta (3.0%), prolonged labor (3.0%), twin pregnancy (3.0%), abortion induced (1.5%), anembryonic gestation (1.5%), breech presentation (1.5%), craniosynostosis (1.5%), fetal malpresentation (1.5%), gestational hypertension (1.5%), fetal hemorrhage (1.5%), hyperemesis gravidarum (1.5%), infantile hemangioma (1.5%), large for dates baby (1.5%), neonatal aspiration (1.5%), premature baby (1.5%), premature delivery (1.5%), rhesus incompatibility (1.5%), and neonatal weight decrease (1.5%) (Figure 5).

Looking at the probability of reporting pregnancy-related adverse events, certolizumab was associated with a lower risk of reporting these events compared to the other anti-TNF drugs (ROR = 0.45, 95% CI = 0.33–0.61). However, when compared specifically to etanercept (ROR = 2.47, 95% CI = 1.57–3.90) and infliximab (ROR = 2.36, 95% CI = 1.50–3.69), certolizumab showed a higher probability.

## 3. Discussion

Pregnant women are typically not allowed to participate in clinical trials, and women who become pregnant during a trial have their treatment stopped immediately [31]. Pregnant women’s complicated physiology, their uncertain willingness to participate, regulations that label them as a “vulnerable” (despite of the more scientifically appropriate “complex”) population in need of special protections in research, and the vague, ambiguous, and restrictive wording of these regulations are some of the reasons why pregnant women participate in research studies [32]. Given this, treatment for pregnant women with psoriasis is limited by this lack of information. Specifically for anti-TNF drugs, experience with these drugs during pregnancy is limited to a small number of cases [21,22,23,33,34]. However, the marketing authorization holders (MAHs) consider that sufficient exposure data have been collected and no longer consider exposure during pregnancy as missing information [8,9,10,11,12]. Our study aimed to evaluate the safety profile of anti-TNF drugs in pregnant women with psoriasis, with a focus on certolizumab, which is considered safer than adalimumab, etanercept, infliximab, and golimumab [33]. We extracted more than 340,000 ICSRs including an anti-TNF drug as a suspected drug and collected in EudraVigilance from 2009 to 2023. Among these, only 1050 ICSRs (0.3%) described cases of pregnant women with psoriasis treated with an anti-TNF drug. There are a higher number of reports for certolizumab than for any other anti-TNF drugs (almost 25% of ICSRs included certolizumab as a suspected drug). Given the patients’ shared characteristics, we assumed that these frequencies can be read as a proxy of trend of use in such population considering the assumed safer profile of certolizumab when used among pregnant women [35]. In fact, given that guidelines, recommendations, and the SmPC all imply that certolizumab may be better than other anti-TNF drugs, due to its structural differences and the lack of a fragment of crystallizable (Fc) portion, therefore, we would actually expect to see a higher use of this medication among pregnant psoriasis patients [13,14]. Moreover, we observed 178 ICSRs of golimumab use with the indication of psoriasis, an indication that is not approved in Europe [12]. This can be attributed to therapeutic errors or to potential miscoding of the appropriate Preferred Term (PT) for the indication (for instance, coding “psoriasis” instead of “psoriatic arthritis”).

More than 80% of the reporters were patients. This is unusual with what is generally seen with spontaneous pharmacovigilance system, where healthcare professionals are approximately 7-fold more involved in reporting suspected ADRs than non-healthcare professionals [36]. It should be noted that our data contained ICSRs for pregnant women, a special population that is particularly vulnerable and empowered in recognizing and reporting events that occur following treatment exposure. This trend, leading to over-reporting, has already been observed among non-healthcare professionals during the COVID-19 vaccination campaign.

In fact, looking at the general population, the non-healthcare/healthcare professionals ratio indicates a reporting rate of 1:1 in the case etanercept and 1:7 in the case of infliximab (data are available in the EudraVigilance dashboard) [37]. The elevated number of ICSRs reported by non-healthcare professionals may compromise the completeness of the data, as suggested by the frequency of unavailable ages (more than half of the ICSRs lacked patient age information). The lack of completeness, as well as the under-reporting, is a prevailing problem in pharmacovigilance, especially in ICSRs reported by non-healthcare professionals [38]. However, in contrast to ICSRs reported by healthcare professionals, patient reports have the potential ability to detect unknown signals since they provide different categories of suspected ADRs for different types of medicines [39,40].

When comparing non-pregnant populations, we observed both similarities and differences in AEs related to certolizumab and other anti-TNF-alpha therapies. Specifically, there were no statistically significant differences between treatments in terms of cardiac, hepatobiliary, metabolic, neoplastic, respiratory, vascular, and reproductive system events. However, certolizumab was associated with a higher risk of gastrointestinal events, immune system disorders, and renal events in non-pregnant women, while these differences were not observed in pregnant women. The risk of cardiovascular risk following the use of anti-TNF drugs is widely debated. In fact, the greater inhibition of TNF receptor 2 (TNFR2)—which is a cardioprotective receptor—by anti-TNF drugs may be the cause of the cardiovascular risks linked to anti-TNF drugs [41]. However, chronic arthritis itself is often characterized by accelerated atherosclerosis and an increased risk of cardiovascular disease [42]. While many potential adverse events of anti-TNF drugs were recorded in pre-marketing trials, several rarer adverse events were recognized as the availability of these drugs increased. For example, hepatic events and respiratory infections were observed in post-marketing studies [43,44]. Considering the pathophysiology of these kinds of adverse events, we did not expect differences between treatment, neither in pregnant patients nor in non-pregnant female patients. In our opinion, the higher risk of gastrointestinal events, immune system disorders, and renal events in non-pregnant women using certolizumab, compared to pregnant women, may be explained by the fact that pregnant patients are more prone to report unknown events and those related to pregnancy outcomes. In fact, as expected, only adverse events classified as congenital, familial, and genetic disorders were more commonly associated with certolizumab in pregnant patients.

Considering both maternal/neonatal and non-maternal/neonatal events, we observed uniformity in the nature of adverse events; each SOC is proportionally represented across all anti-TNF drugs without significant variability. These results are not consistent with the current pharmacovigilance evidence from the general population, which showed that each anti-TNF drug is associated with a distinct profile of adverse events [45]. This implies that pregnancy could be a factor influencing both the type of reported adverse events and consequently their outcomes. However, the currently available pharmacovigilance studies are not directly comparable to ours due to the inherent differences in the data sources.

Considering the pregnancy outcomes, spontaneous abortion is the most reported suspected ADR, predominantly followed by the use of adalimumab and certolizumab. Moreover, we observed other adverse pregnancy outcomes such as fetal death, premature birth, and fetal distress syndrome. It should be highlighted that our analysis was strictly based on data from spontaneous pharmacovigilance reports, and we are unable to establish a causal relationship between the treatment and the event. Confounding factors such as psoriasis itself should be considered. Specifically, the risk of negative pregnancy outcomes may increase due to the progression of psoriasis infection and psoriasis-related comorbidities such as diabetes, cardiovascular disease, and depression [46,47,48,49,50,51,52,53,54]. Nevertheless, the evidence is more limited and conflicting. A meta-analysis published in 2021 has summarized the evidence from observational studies that studied the maternal outcomes (e.g., spontaneous abortion, caesarean delivery, preterm birth, preterm birth, gestational diabetes, gestational hypertension, and ante- or postpartum hemorrhage) and/or neonatal outcomes (e.g., congenital malformations, low birth weight, and neonatal mortality) in pregnant women with psoriasis or psoriatic arthritis compared with healthy subjects [55]. This study found substantial evidence that maternal outcomes are negatively affected in women with psoriasis and psoriatic arthritis, but there is no increased risk of negative neonatal outcomes.

In line with expectations based on preclinical study findings, we observed that certolizumab was associated with a lower risk of reporting pregnancy-related adverse events when compared to all the other anti-TNF drugs. However, results from the disproportionality analysis are conflicting; in fact, when compared specifically to etanercept and infliximab, certolizumab showed a higher probability of reporting these events.

We observed a similarity in terms of safety in pregnant patients with psoriasis among all the anti-TNF-alpha drugs analyzed. In our opinion, the current recommendations regarding the use of anti-TNF drugs in pregnant women, which suggest a better safety profile for certolizumab, are based on partial evidence. Specifically, the current recommendations are based on results from preclinical studies that support the hypothesis that certolizumab has low placental transfer in pregnant women given that it lacks a fragment of crystallizable portion, which is a part of a complete antibody [15,17,18,19]. Moreover, observational post-marketing studies do not allow to draw a definitive conclusion [55].

We advocate for further real-world analyses to ascertain or challenge certolizumab’s actual better profile in pregnant women with psoriasis.

### Strengths and Limitations

The key strengths of our study is the use of data from one of the largest of spontaneous reports [37]. It covers heterogeneous information from different countries and populations and is widely employed for the identification of potential risks and emerging questions, including in specific subpopulations such as children and pregnant women [56,57]. In fact, the rationale for pharmacovigilance is based on the need to address the limits of pre-marketing trials: small sample sizes, limited duration, and the exclusion or the underrepresentation of specific population groups [58,59].

On the other hand, our study has several limitations. Identifying pregnancy-related reports within spontaneous reporting databases, such as EudraVigilance, posed significant challenges in our analysis. The existing literature underscores the absence of a standardized algorithm endorsed by regulatory authorities for systematically flagging reports involving pregnant women [60]. In our study, we included only five Preferred Terms strictly associated with pregnancy. While this strategy aimed to ensure accuracy in identifying true pregnancy cases within the EudraVigialance database, the stringent criteria for term inclusion may have inadvertently excluded reports related to pregnant patients. Moreover, our results are subject to typical biases encountered in pharmacovigilance, including under-reporting, the tendency for serious events to be reported more frequently than non-serious events, and notoriety bias (the spontaneous reporting may be influenced in unknown ways by safety alerts) [61]. Moreover, the analyzed ICSRs were predominantly patient-reported (in contrast to typical pharmacovigilance practices), which may have influenced the quality and completeness of the ICSRs themselves. Additionally, since spontaneous data reporting does not include the total number of drug users or accurately quantify risks, Supplementary Data from other sources are required to assess risk comprehensively. Finally, given the nature of the analyzed data, we cannot infer causality, particularly considering the presence of strong confounding factors such as psoriasis itself.

## 4. Materials and Methods

### 4.1. Data Source

Data on Individual Case Safety Reports (ICSRs) of suspected adverse drug reactions (ADRs) by the anti-TNF drugs infliximab, etanercept, adalimumab, certolizumab, and golimumab were retrieved from the EudraVigilance (EV) database [37]. We selected all ICSRs reported in EV from 1 January 2009 to 31 December 2023. This decision was made to ensure the same temporal range between the selected drugs, given that the last anti-TNF drugs to be approved, certolizumab and golimumab, were both approved in 2009 [11,12]. Although golimumab is not approved for the treatment of psoriasis in Europe, we have decided to include it in our analysis due to the possibility of its administration either by mistake or as an off-label use. Additionally, the indication “Psoriasis” itself does not unequivocally confirm a diagnosis of psoriasis, as there may be potential miscoding issues related to the Preferred Term (PT).

We considered the following information from each ICSR: a unique identifier number, the report type (spontaneous or non-spontaneous), the gateway receipt date, the primary source qualification (healthcare or non-healthcare professional), the primary source country (European or non-European Economic Area), the patient age group, the patient sex, and a list of suspected ADRs including their duration, outcome, and seriousness. The reported suspected ADRs are coded as PT according to the Medical Dictionary for Regulatory Activities (MedDRA) [62]. Each PT is a distinct medical concept for a symptom, sign, disease diagnosis, therapeutic indication, investigation, surgical or medical procedure, and medical social or family history characteristic. According to the MedDRA hierarchical structure, PTs are grouped into System Organ Classes (SOCs), which are grouped based on etiology, manifestation site, or purpose. Regarding the seriousness, an event is defined as serious when the ADR results in death, is life-threatening, requires/prolongs hospitalization, results in persistent or significant disability/incapacity, is a congenital anomaly/birth defect, or results in some other clinically important condition. Moreover, the outcomes may be labeled as “Recovered/Resolved”, “Recovering/Resolving”, “Recovered/Resolved With Sequelae”, “Not Recovered/Not Resolved”, “Fatal”, and “Unknown”. Finally, the ICSRs include data on suspected and concomitant medications. A suspected drug is characterized as a medication potentially linked to the observed suspected ADR, whereas a concomitant medication refers to a drug the patient is exposed to, which may not necessarily be associated with the suspected ADR. For each one, administration route, therapy duration, dosages, and therapeutic indication were reported.

### 4.2. Study Population

We selected only the ICSRs involving patients with psoriasis treated with an anti-TNF-alpha, excluding cases with other indications (e.g., rheumatoid arthritis, juvenile idiopathic arthritis, Crohn disease, and psoriatic arthritis) or unspecified indications. Moreover, given our aim in analyzing the safety of anti-TNF drugs in pregnant women, we identified cases involving pregnant women by filtering the initial dataset by the following PTs: “Exposure during pregnancy”, “Maternal exposure during pregnancy”, “Pregnancy”, “Second trimester pregnancy”, and “Third trimester pregnancy”.

### 4.3. Descriptive Analysis

We conducted a descriptive analysis of the demographic characteristics of ICSRs and the key features of the suspected ADRs, stratifying the results by each anti-TNF drug. We described the frequencies of suspected ADRs by SOC. Moreover, we focused our analysis on the specific pregnancy outcomes and fetal/neonatal disorders—the list of PTs utilized for the analysis is provided in the Appendix A. Finally, we explored the outcomes associated with neonatal infections, including ICSRs with all PTs reported in the High-Level Term (HLT) “Neonatal infections (excluding congenital infections)”.

### 4.4. Disproportionality Analysis

A disproportionality analysis was conducted in order to estimate the odds of reporting specific adverse events (classified by SOC) associated with certolizumab compared to other anti-TNF drugs, with subsequent comparison of reporting odds ratio (ROR) with a 95% confidence interval (95% CI) by comparing certolizumab with the other anti-TNF drugs, both in pregnant (i.e., our study population) and non-pregnant patients used as a control group. Specifically, we calculated the ROR with 95% CI as follow:$$ROR=\frac{a/b}{c/d},\;95\%\;CI=exp\left\{\ln(ROR)\left.\pm 1.96\sqrt{\frac{1}{a}+\frac{1}{b}+\frac{1}{c}+\frac{1}{d}}\right\}\right.$$
where (a) was the number of events related to a specific SOC when certolizumab was the suspected drug, (b) was the number of events not related to that specific SOC when certolizumab was the suspected drug, (c) was the number of events related to a specific SOC when an anti-TNF drug (excluding certolizumab) was the suspected drug, and (d) was the number of events not related to that specific SOC when an anti-TNF drug (excluding certolizumab) was the suspected drug. We did not calculate the ROR when (a) or (c) were less than three. A *p* value of <0.05 was used for statistical significance.

Moreover, we focused our analysis on the pregnancy-related adverse events, determining whether pregnant women with psoriasis exhibit a lower/higher probability of reporting pregnancy-related adverse events when using certolizumab compared to other anti-TNF drugs. In this case, (a) was the number of adverse pregnancy outcomes and fetal/neonatal disorders associated with certolizumab as a suspected drug, (b) was the number of adverse events not related to adverse pregnancy outcomes and fetal/neonatal disorders associated with certolizumab as a suspected drug, (c) was the number of adverse pregnancy outcomes and fetal/neonatal disorders associated with an anti-TNF drug (excluding certolizumab) as a suspected drug, and (d) was the number of adverse events not related to adverse pregnancy outcomes and fetal/neonatal disorders associated with an anti-TNF drug as a suspected drug.

### 4.5. Sensitivity Analysis

To address the potential impact of missing indications in the EV data, we conducted a sensitivity analysis by including ICSRs where the indication for anti-TNF-alpha drugs was not specified.

Data management and all statistical analysis were performed using the R Statistical Software (version 4.4.0; R Foundation for Statistical Computing, Wien, Austria).

## 5. Conclusions

Our analysis provided an overview of spontaneous reports of suspected adverse drug reactions occurring in pregnant women with psoriasis treated with certolizumab, adalimumab, etanercept, infliximab, or golimumab. In particular, we focused our analysis on the maternal and neonatal outcomes in order to verify the actual superiority of certolizumab, as demonstrated in preclinical studies. The findings from our study depicted a safety profile that overlapped between each anti-TNF drug, both in maternal/neonatal outcomes and other adverse events. Given the nature of our pharmacovigilance study, we refrain from drawing definitive conclusions. However, due to the limited evidence regarding the safety profile of anti-TNF drugs used during pregnancy (as well as in other special population, including one of children), we advocate for further investigations before making concrete recommendations.

## Figures and Tables

**Figure 1 pharmaceuticals-17-00904-f001:**
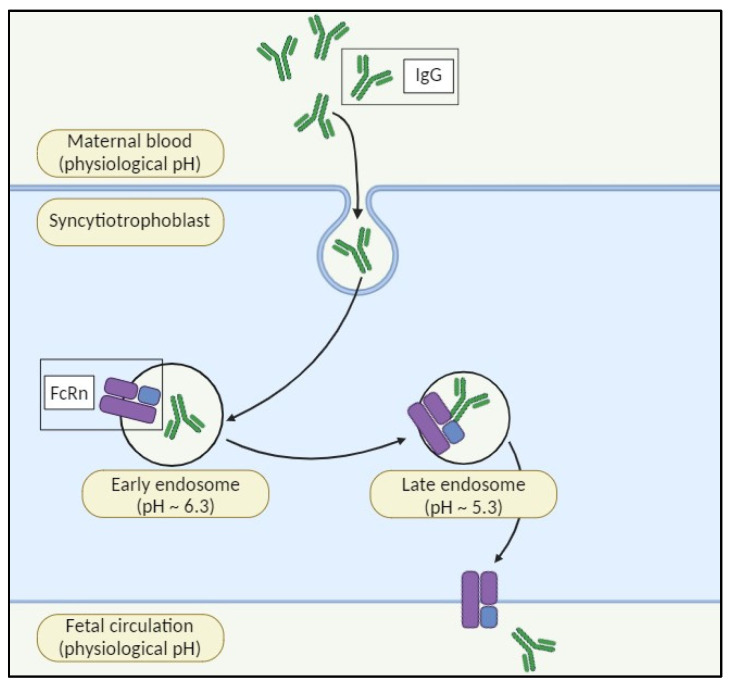
The neonatal Fc receptor (FcRn) for immunoglobulin G (IgG) plays a crucial role in the perinatal transfer of IgG. Maternal IgG is transferred to the fetus via the syncytiotrophoblast of the placenta. FcRn is located in the internal vesicles of the syncytiotrophoblast. Upon acidification within the endosome, FcRn binds to maternal IgG and transcytoses it to the fetal circulation, where it is released at the physiological pH.

**Figure 2 pharmaceuticals-17-00904-f002:**
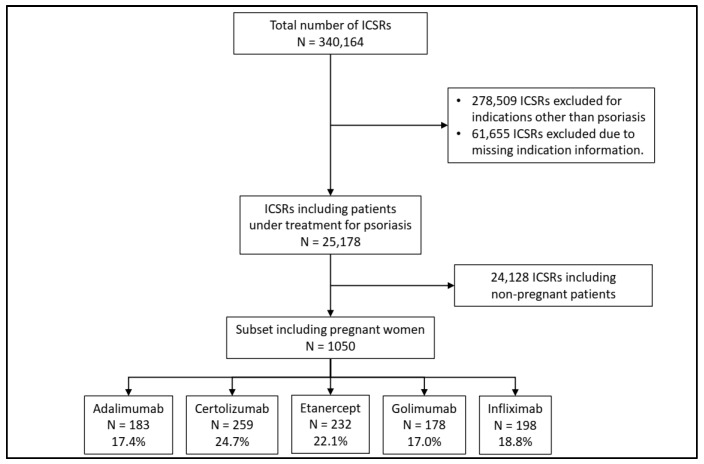
Flowchart for the inclusion and exclusion criteria of Individual Case Safety Reports (ICSRs).

**Figure 3 pharmaceuticals-17-00904-f003:**
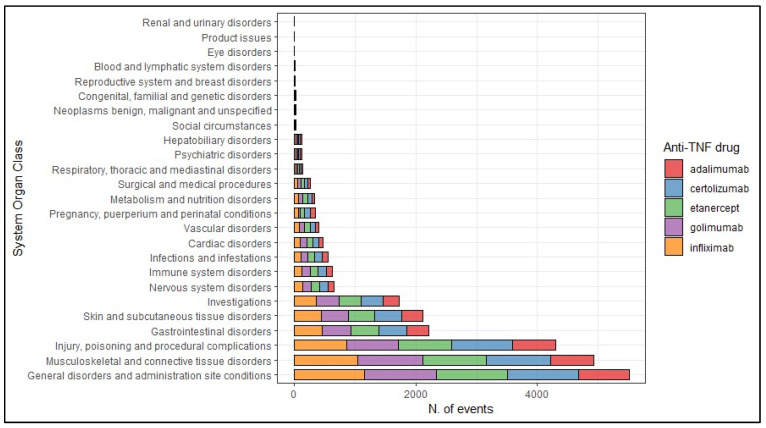
Frequency of the suspected adverse drug reactions according to their System Organ Class.

**Figure 4 pharmaceuticals-17-00904-f004:**
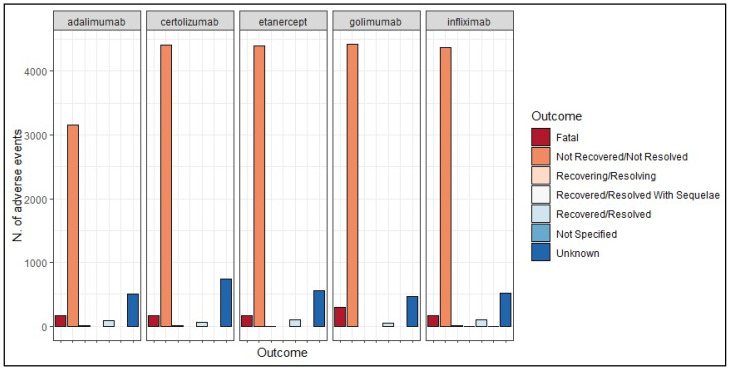
Frequency of the suspected adverse drug reactions (both pregnancy- and non-pregnancy-related reactions) according to their outcomes.

**Figure 5 pharmaceuticals-17-00904-f005:**
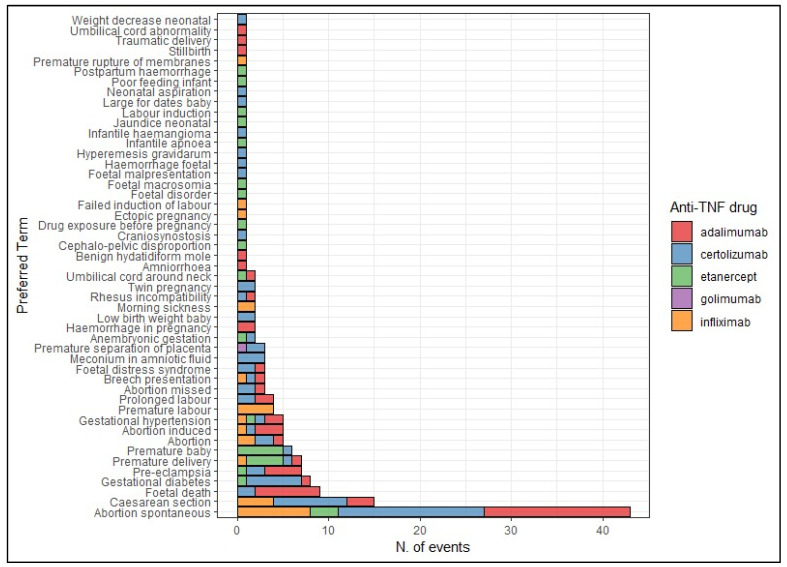
Distribution of suspected adverse drug reactions included in the SMQs “Pregnancy, labor and delivery complications”, “Fetal disorders”, “Neonatal disorders”, and “Termination of pregnancy”.

**Table 1 pharmaceuticals-17-00904-t001:** Characteristics of Individual Case Safety Reports (ICSRs) included in the analysis.

	Anti-TNF Drugs				
	Adalimumab	Certolizumab	Etanercept	Golimumab	Infliximab
**Number of ICSRs**	183	259	232	178	198
**Age range (%)**					
0–1 month	1 (0.5)	3 (1.2)	1 (0.4)	0 (0.0)	0 (0.0)
2 months–2 years	0 (0.0)	1 (0.4)	0 (0.0)	0 (0.0)	0 (0.0)
3–11 years	0 (0.0)	0 (0.0)	0 (0.0)	0 (0.0)	0 (0.0)
12–17 years	0 (0.0)	0 (0.0)	1 (0.4)	0 (0.0)	0 (0.0)
18–64 years	90 (49.2)	105 (40.5)	97 (41.8)	62 (34.8)	77 (38.9)
Not available	92 (50.3)	150 (57.9)	133 (57.3)	116 (65.2)	121 (61.1)
**Type of report (%)**					
Spontaneous	92 (50.3)	150 (57.9)	133 (57.3)	116 (65.2)	121 (61.1)
Non-spontaneous	91 (49.7)	109 (42.1)	99 (42.7)	62 (34.8)	77 (38.9)
**Source qualification** = Patient (%)	155 (84.7)	217 (83.8)	207 (89.2)	176 (98.9)	191 (96.5)

**Table 2 pharmaceuticals-17-00904-t002:** Comparison of adverse events between pregnant and non-pregnant women by comparing the odds of reporting events with certolizumab versus other anti-TNF drugs.

	Certolizumab vs. Other TNF-Alpha Drugs (ROR (95% CI))	
System Organ Class	Pregnant Population	Non-Pregnant Population
Blood and lymphatic system disorders	(less than 3 cases)	1.33 (0.75–2.37)
Cardiac disorders	1.00 (0.80–1.25)	1.13 (0.69–1.86)
Congenital, familial, and genetic disorders	2.07 (0.87–4.94)	(less than 3 cases)
Ear and labyrinth disorders	(less than 3 cases)	1.91 (0.98–3.73)
Endocrine disorders	(less than 3 cases)	3.27 (1.51–7.08)
Eye disorders	(less than 3 cases)	2.05 (1.41–2.98)
Gastrointestinal disorders	0.94 (0.84–1.05)	2.30 (1.93–2.74)
General disorders	0.97 (0.90–1.04)	2.88 (2.60–3.19)
Hepatobiliary disorders	0.82 (0.52–1.30)	0.76 (0.39–1.48)
Immune system disorders	1.01 (0.83–1.22)	3.53 (2.66–4.68)
Infections and infestations	1.00 (0.82–1.23)	2.05 (1.75–2.40)
Injury, poisoning, and procedural complications	1.12 (1.03–1.21)	5.40 (4.72–6.18)
Investigations	0.94 (0.83–1.06)	1.25 (1.01–1.56)
Metabolism and nutrition disorders	0.99 (0.77–1.29)	1.28 (0.80–2.04)
Musculoskeletal and connective tissue disorders	0.97 (0.90–1.05)	2.94 (2.60–3.34)
Neoplasms benign, malignant, and unspecified	1.03 (0.42–2.56)	0.52 (0.34–0.78)
Nervous system disorders	0.95 (0.79–1.15)	1.89 (1.57–2.28)
Pregnancy, puerperium, and perinatal conditions	1.30 (1.02–1.65)	2.08 (0.98–4.42)
Product issues	(less than 3 cases)	3.29 (2.18–4.96)
Psychiatric disorders	0.96 (0.63–1.47)	1.87 (1.41–2.48)
Renal and urinary disorders	(less than 3 cases)	2.49 (1.59–3.92)
Reproductive system and breast disorders	2.41 (0.68–8.56)	1.38 (0.71–2.69)
Respiratory, thoracic, and mediastinal disorders	1.05 (0.70–1.59)	1.26 (0.94–1.69)
Skin and subcutaneous tissue disorders	0.98 (0.88–1.09)	3.25 (2.90–3.63)
Social circumstances	1.45 (0.64–3.29)	4.56 (2.85–7.30)
Surgical and medical procedures	1.15 (0.87–1.52)	8.25 (6.39–10.65)
Vascular disorders	0.99 (0.78–1.27)	1.50 (0.99–2.27)

## Data Availability

Dataset is available on request from the authors.

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
