# Peer review of "Pregnancy Recommendations Solely Based on Preclinical Evidence Should Be Integrated with Real-World Evidence: A Disproportionality Analysis of Certolizumab and Other TNF-Alpha Inhibitors Used in Pregnant Patients with Psoriasis"

_pharmaceuticals, 2024, doi:10.3390/ph17070904_

Round 1

Reviewer 1 Report (Previous Reviewer 1)

Comments and Suggestions for Authors

The authors answered to the addressed issues and updated the manuscript; however no document showing the track of answers was found.

Author Response

Thank you again for your time and effort in reviewing our manuscript. The previous revisions have been highlighted within the text as requested, and the new modifications made to the manuscript have all been tracked for clarity.

We greatly appreciate your contribution.

Reviewer 2 Report (Previous Reviewer 2)

Comments and Suggestions for Authors

The authors revised the manuscript by considering the reviewers' comments, hereby I endorse the manuscript for publication in its present form. 

Author Response

Thank you again for your time and effort in reviewing our manuscript.

Reviewer 3 Report (New Reviewer)

Comments and Suggestions for Authors

Dear authors, 

I read with interest your manuscript, which aims to look at the safety of TNF-alpha in pregnant women. I have some comments:

Title: I don't think the "comparative pharmacovigilance study" expression is correct. Maybe "disproportionality analysis" is more appropriate.

Methods:
- Golimumab is not approved for psoriasis in Europe (only psoriatic arthritis). Please motivate the choice to include this drug since EUDRAVIGILANCE is the European spontaneous reporting system.

Results:
- you found more than 170 ICSRs related to golimumab in psoriasis (not approved). Please try to explain this result.

- An overall description (as done in Table 1) for all other records regarding TNF-alpha not related to pregnancy should be presented, for instance, in the Supplementary material since a disproportionality analysis is conducted also on such reports.

- it is important to understand the impact of possible missing indications in EUDRAVigilance. In Figure 2 (Flow chart) authors should specifically report those records with an indication different respect to psoriasis and those with a missing one. To understand the extent of the missing indication, I would also suggest a sensitivity analysis by including those missing records in the disproportionality analysis to be reported in the Supplementary material.

-Table 1: Since the patients included were mostly pregnant women, i think a more detailed stratification about patients aged 18+ is recommended.

Introduction/Discussion

- When explaining the rationale of the study (i.e. starting from line 115 to 127) authors should also mention some European data about the different characteristics of patients using TNF-alpha in the real-world vs those included in randomized clinical trials. For example, among patients with psoriasis, certolizumab was most used by women in real-world respect to the clinical setting (https://doi.org/10.1016/j.phrs.2024.107074), thus limiting the generalizability of the RCT results

- Maybe the introduction can be shorter and more concise regarding the preclinical part

- Please check this reference (https://ncbi.nlm.nih.gov/pmc/articles/PMC9684212/) about identifying reports related to pregnancy in the spontaneous reporting system. Please discuss in the limitation section the different strategy used.

Author Response

Thank you very much for your feedback and invaluable suggestions. We sincerely appreciate your thoughtful insights into our manuscript. Following your suggestions, we have made the following changes:

- We have revised the title as suggested.

- We have provided justification for the inclusion of golimumab in both the Methods (from line 379 to 383) and the Discussion (from line 254 to 257) sections.

- We presented an overall description (as done in Table 1) for all other records regarding TNF-alpha not related to pregnancy (Supplementary table 2)

- We have revised the flowchart (Figure 2) to include specific information on the number of ICSRs with missing indications, as well as those with indications different from psoriasis. Additionally, as suggested, we conducted a sensitivity analysis. The results of this analysis are detailed in the Results section (from line 192 to 205) and the Methods section (from line 451 to 453). Furthermore, we have included a supplementary table providing comprehensive data on the missing indications and the results of the sensitivity analysis.

- We appreciate your suggestion regarding the stratification of patients aged 18+. However, due to limitations in our access to complete data, we were unable to provide a more detailed stratification beyond what is presented in Table 1.

- We have incorporated the European data about the characteristics of patients using TNF-alpha inhibitors in real-world settings compared to those in randomized clinical trials. Specifically, we have mentioned the findings regarding certolizumab and its usage patterns among women with psoriasis, highlighting the potential impact on the generalizability of RCT results (from line 123 to 128).

- In the limitations section of our manuscript, we have now included a discussion on the challenges associated with identifying reports related to pregnancy (from line 352 to 360).

Round 2

Reviewer 3 Report (New Reviewer)

Comments and Suggestions for Authors

The authors replied to all my comments

This manuscript is a resubmission of an earlier submission. The following is a list of the peer review reports and author responses from that submission.

Round 1

Reviewer 1 Report

Comments and Suggestions for Authors

The authors performed a pharmacovigilance study on the effect of drugs such as certolizumab on the pregnant women with psoriasis. The authors used as data, the statistics provided by European Medicines Agency, online available.

The authors concluded that “This is the first pharmacovigilance study of anti-TNF drugs used in pregnant women  with psoriasis.” However, it was found a recent study with similar outcome (doi: 10.1177/1759720X221087650)

The study is important as it is stressing out a major issue nowadays; however, the authors did not mention how many  pregnant ladies (as percentages) have the diagnosis of psoriasis; this statistic will be useful to support the aim of the present study.

Reviewer 2 Report

Comments and Suggestions for Authors

Pregnancy-induced immune response is highly regulated to protect the fetus, any deregulations in the immune response affect the fetus or cause fetus abnormalities. There have been very limited drugs that are safe in the pregnancy condition, and observation is needed throughout the period. Pharmacovigilance observations are vital to any drug product to report the unwanted effects in the host. Gaio et al., have submitted the Pregnancy recommendations solely based on preclinical evidence that should be integrated by real-world evidence: a comparative pharmacovigilance study of certolizumab and other TNF alpha inhibitors used in pregnant patients with psoriasis.”.

As mentioned by the authors this is a descriptive comparative analysis of the anti-TNF drugs in pregnant patients with psoriasis. The main focus was on certolizumab.

Comments

My first impression of the read is it would be suitable for short communication rather than full-length articles, due to descriptive analysis.

The manuscript is lacking figures, which should be included wherever possible, e.g., anti-TNF drugs mechanisms in pregnant patients with psoriasis.

With regards to certolizumab, the authors have not covered how the adverse events are happening, despite the lack of an Fc portion.